# High density lithium niobate photonic integrated circuits

Zihan Li[1,2,4], Rui Ning Wang [1,2,4], Grigory Lihachev[1,2,4], Junyin Zhang [1,2], Zelin Tan[1,2], Mikhail Churaev [1,2], Nikolai Kuznetsov [1,2], Anat Siddharth [1,2], Mohammad J. Bereyhi [1,2,3], Johann Riemensberger [1,2] & Tobias J. Kippenberg [1,2] ✉

Photonic integrated circuits have the potential to pervade into multiple applications traditionally limited to bulk optics. Of particular interest for new applications are ferroelectrics such as Lithium Niobate, which exhibit a large Pockels effect, but are difficult to process via dry etching. Here we demonstrate that diamond-like carbon (DLC) is a superior material for the manufacturing of photonic integrated circuits based on ferroelectrics, specifically LiNbO$_3$. Using DLC as a hard mask, we demonstrate the fabrication of deeply etched, tightly confining, low loss waveguides with losses as low as 4 dB/m. In contrast to widely employed ridge waveguides, this approach benefits from a more than one order of magnitude higher area integration density while maintaining efficient electro-optical modulation, low loss, and offering a route for efficient optical fiber interfaces. As a proof of concept, we demonstrate a III-V/LiNbO$_3$ based laser with sub-kHz intrinsic linewidth and tuning rate of 0.7 PHz/s with excellent linearity and CMOS-compatible driving voltage. We also demonstrated a MZM modulator with a 1.73 cm length and a halfwave voltage of 1.94 V.

Photonic integrated circuits (PICs) based on silicon (Si) have transitioned from academic research to use in data centers over the past two decades[1,2]. In the recent wave of development, silicon nitride—initially motivated by its nonlinear properties for microcomb generation[3,4]—has emerged as an integrated photonics platform in itself, offering low loss at the dB-meter level, nonlinear operation, high power handling capability, a wide optical transparency window and space compatibility. It has allowed novel capabilities including chip-scale optical frequency comb sources[5], traveling-wave optical parametric amplifiers[6,7], Erbium waveguide amplifiers[8], integrated lasers that operate in the visible and near UV spectral range[9], and is challenging the best bulk legacy lasers in phase noise and tuning[10,11]. The commercial availability of lithium niobate on insulator (LNOI)—and ferroelectric thin film materials on insulator[12,13] in general—can extend the functionality further by offering one of the highest Pockels

coefficients, required to realize volt level high-speed modulators[14,15], electro-optical frequency combs[16], photonic switching networks[17], delay lines[18], on-chip broadband spectrometers[19] and lasers[20]. Periodic poling of thin film-based LiNbO$_3$ ridge waveguides has allowed on-chip frequency doublers[21,22], squeezed light sources[23] and optical parametric oscillators[24]. LiNbO$_3$ also features large second-order nonlinear susceptibility for optical frequency conversion[25] and piezoelectric coefficient enabling advanced on-chip acousto-optics[26]. Integrated photonic circuits critically rely on achieving wafer-scale manufacturing, which also exhibits low loss and attains lithographic precision and reproducibility. A critical manufacturing step is the etching process, which transfers the lithographic pattern into the photonic device layer. While for currently employed foundry-compatible photonic material platforms, in particular, silicon or silicon nitride, mature processing is available, the latter cannot be readily extended to the rapidly emerging

[1]Institute of Physics, Swiss Federal Institute of Technology Lausanne (EPFL), CH-1015 Lausanne, Switzerland. [2]Center of Quantum Science and Engineering (EPFL), CH-1015 Lausanne, Switzerland. [3]Luxtelligence SA, CH-1015 Lausanne, Switzerland. [4]These authors contributed equally: Zihan Li, Rui Ning Wang, Grigory Lihachev. ✉e-mail: tobias.kippenberg@epfl.ch

platforms based on ferroelectric materials, such as lithium niobate on insulator[27]. Direct etching of lithium niobate is usually based on argon ion bombardment, which is a strong physical process that cannot achieve a high etch selectivity between lithium niobate and common hard mask materials, such as $SiO_2$ and a-Si (see comparison of different dry etching methods in Supplementary Fig. 2). As a result, state of the art in $LiNbO_3$ integrated photonics technology[28] are shallow ridge waveguides with close to unity ridge to slab ratios and strongly slanted sidewalls ($\approx 60°$). The thick slab entails strong limitations in terms of waveguide bending radius as well as requiring more challenging process control due to the partial etch. This complicates efforts to establish and qualify process design kits for $LiNbO_3$ photonic integrated waveguide platforms. Metallic masks, such as chromium[29], feature high etch selectivity but are polycrystalline with rough sidewalls and optically absorbent etch product redeposition. Heterogeneous integration of $LiNbO_3$ with silicon or silicon nitride integrated photonic circuits is another approach. Such hybrid waveguides have been successfully achieved[30,31], with excellent low loss performance, yet exhibit additional complexity due to the use of wafer bonding onto pre-fabricated substrates, and requiring tapers for mitigating losses from transitioning into and out of the bonded areas. Hence a $LiNbO_3$ photonic integrated circuit platform featuring ultra-low loss fully etched strip waveguides with vertical sidewalls, already well established and successfully commercially employed for silicon and silicon nitride, would be highly desirable. Here we overcome this challenge by introducing DLC, diamond-like-carbon, as a hard mask process, and demonstrate $LiNbO_3$ strip waveguide based photonic integrated circuits with high yield, ultra-low loss (<4 dB/m) that benefit from the 10-fold higher area density that previous ridge, i.e., partially etched, approaches.

## Results

DLC was discovered in 1953[32] as a byproduct of a carburization experiment by H.P.P. Schmellenmeier, who obtained a black reflective film on a metal alloy in a glow-discharge plasma of $C_2H_2$, which featured high hardness, scratch resistance, and amorphous X-ray scattering characteristics. The material due to the presence of $sp^3$ i.e., diamond-like bonds of carbon, exhibits some of the extreme properties of diamond, in terms of hardness, elastic modulus, and chemical inertness, but is amorphous and can be deposited at low cost as a thin film. Today, DLC is ubiquitously used as a protective coating[33,34] for magnetic read heads[35], medical devices, razor blades or MEMS

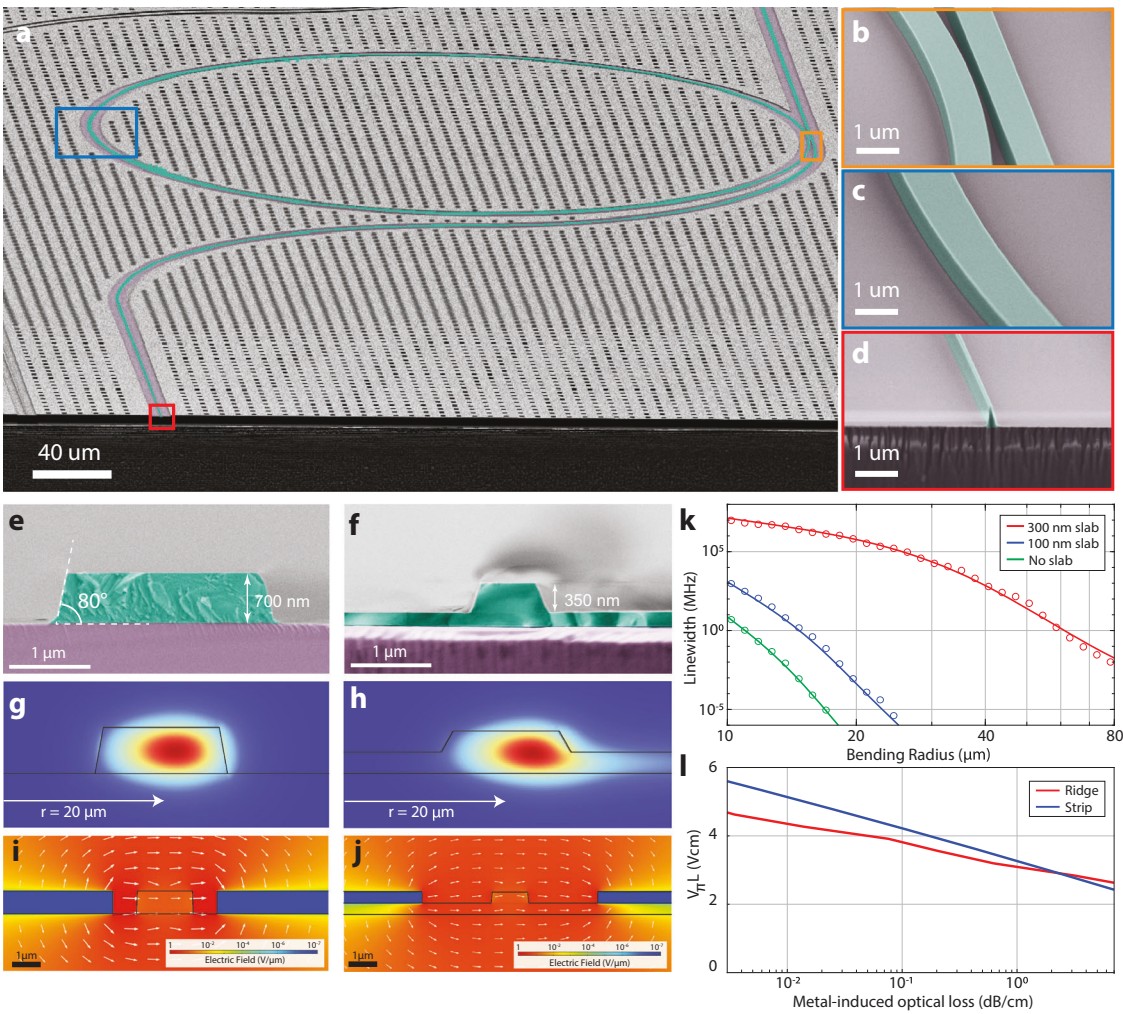

**Fig. 1 | Tightly confining lithium niobate photonic integrated circuit platform.** **a** Scanning electron micrograph (SEM) of a $LiNbO_3$-based photonic integrated circuit with high resolution insets of microring coupling section ((**b**, orange), curved waveguide (**c**, blue), and inverse taper for fiber coupling (**d**, red)). **e** SEM of a fully etched optical waveguide with cross section 2 μm × 0.7 μm. **f** SEM of a partially etched ridge waveguide 1.5 μm × 0.6 μm, 350 nm etch depth and 250 nm $LiNbO_3$ slab. **g**, **h** Optical mode field distribution of curved waveguides corresponding to (**e**, **f**) with 20 μm bend radius. **i**, **j** Electrical field distribution in log-scale for slab and ridge waveguides with electrode spacing 0.75 μm and 2.5 μm, which are selected to feature equal metal-induced optical losses. **k** FEM simulations of LN waveguide bending loss for different slab heights and waveguide cross-section 2 μm × 0.6 μm. **l** FEM simulation of the voltage-length product of electro-optical modulation for strip and ridge waveguides as a function of metal-induced waveguide loss.

sensors[36,37]. DLC films can be deposited by plasma-enhanced chemical vapor deposition (PECVD) with hardness up to 20 GPa or physical vapor deposition with hardness up to 80 GPa[38]. As an amorphous carbon allotrope, DLC can be readily etched with oxygen plasma which gives rise to a smooth boundary after etching. It has been recognized already in the 1990s that DLC is also suitable as etch masks for integrated circuit fabrication[39], yet its use as the hard mask has not proliferated. The low argon ion sputtering yield[40] and excellent chemical stability of DLC are much desired for the etching of any integrated optical materials such as LiNbO$_3$, SiO$_2$, BaTiO$_3$, Si, and Si$_3$N$_4$ as they limit mask erosion processes that determine surface roughness and lead to slanted waveguide sidewall angles.

## Tightly confining lithium niobate photonic integrated circuit platform

Figure 1a depicts a scanning electron micrograph (SEM) of an optical microresonator manufactured with our DLC-based process. Colored insets highlight the waveguide sidewalls and high aspect ratio positive and negative features such as directional couplers and inverse tapers (see Fig. 1b–d). The critical dimension is 300 nm for positive and 200 nm for negative features with a sidewall angle of 80°. Inverse tapers with 250 nm width expand the waveguide mode sufficiently to support low loss input coupling from a lensed fiber to the LiNbO$_3$ PIC, reaching 3 dB/facet level without the need for a two-step etching process that is required to reach similar levels in ridge waveguides[41] and heterogeneously integrated hybrid LiNbO$_3$-Si$_3$N$_4$ waveguides[30,31] to mitigate the typical 5–10 dB facet insertion loss. Figure 1e, f compares our waveguides with a ridge LiNbO$_3$ waveguide with 350 nm etch depth and 250 nm remaining slab with a top width of 1.5 µm where low-

loss operation has been demonstrated[42]. Crucially, the LiNbO$_3$ slab induces a continuum of leaky modes with a high refractive index that couples to the guided mode in curved waveguide sections. The numerical simulations presented in Fig. 1g, h, k reveal a 9 and 12 order of magnitude reduction of bending losses for curved waveguides with 20 µm radii if the slab is thinned to 100 nm and fully removed, respectively. Similar top waveguide widths of 2 µm and thicknesses of 0.7 µm are assumed in all the numerical simulations. Fully etched strip waveguides (see Fig. 1a) feature tight optical mode confinement at bending radii below 20 µm, enabling dense integration of optical components on chip with a factor of 16 area density improvement over currently employed ridge waveguide structures for devices such as ring resonators. While a residual slab layer can improve the modulation efficiency due to the high dielectric constant LiNbO$_3$, the increased optical confinement allows to place electrodes closer to the waveguide, which enables electro-optical modulation efficiencies that are competitive with ridge waveguides[43] (see Supplementary Fig. 3) and hybrid waveguides[31,44] without inducing excess Ohmic losses at the metal electrode interfaces (see Fig. 1i, j). Indeed our numerical simulations show that only a 10% penalty in DC modulation voltage-length product is incurred by transitioning to a fully etched i.e., strip waveguide at an Ohmic loss target of 0.1 dB m$^{-1}$ (see Fig. 1l).

## Fabrication process with DLC hard mask

Figure 2a depicts the schematic process flow of our low-loss LiNbO$_3$ strip waveguide fabrication. We start the fabrication using commercially available thin film lithium niobate on insulator wafer (NanoLN) with X-cut(600 nm) or Z-cut(700 nm) thick LiNbO$_3$ layer, 4.7 µm buried oxide (thermally grown) on a 525 µm thick Si substrate. The salient

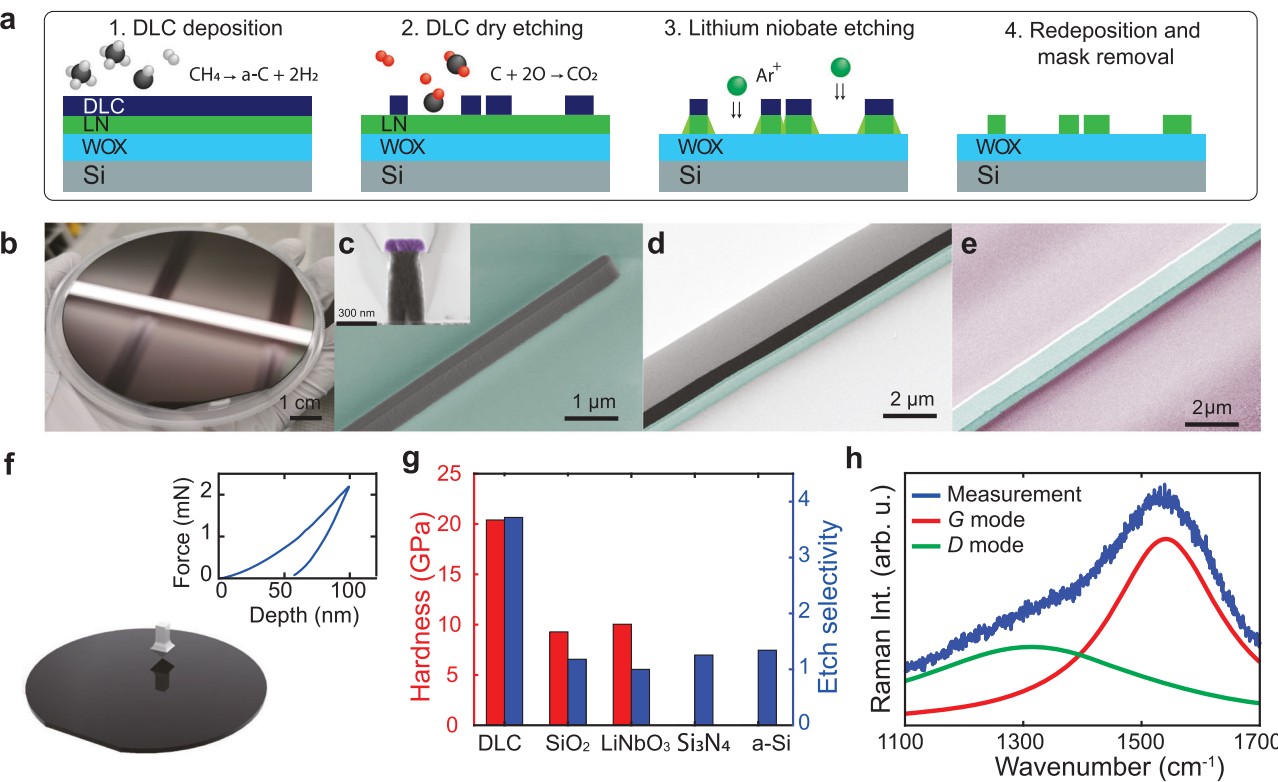

**Fig. 2 | Fabrication process based on diamond-like carbon (DLC) hardmask.** **a** Process flow of the LNOI waveguide fabrication including diamond-like carbon (DLC) hard mask deposition via plasma enhanced chemical vapor deposition (PECVD) from methane precursor, DLC dry etching via oxygen plasma, lithium niobate etching via argon ion beam etching (IBE), followed by redeposition and mask removal. LiNbO$_3$ is illustrated in green, SiO$_2$ in light blue, DLC in dark blue and Si in gray. **b** Photo of a deposited DLC film on the silicon wafer. SEM images of the taper pattern and the cross section with 250 nm width after DLC dry etching (**c**) DLC mask and the LiNbO$_3$ waveguide after redeposition cleaning (**d**) and the LN waveguide with 0.7 µm height and 2 µm width after mask removal (**e**). In the SEM images, the LiNbO$_3$ is colored in green, DLC in black, Si$_3$N$_4$ in purple and SiO$_2$ in light purple. **f** The schematic and the result of indentation hardness measurement for DLC. **g** Hardness and etching selectivity chart for different materials. **h** Raman spectrum and fitting result for DLC film.

feature of our fabrication process is the use of a 300 nm thick DLC film, which is grown via PECVD from a methane precursor, as a hard mask material for the physical ion beam etching process. The diamond indentation hardness of the as-deposited DLC film is 19–23 MPa (see Fig. 2f and Supplementary Fig. 1), which is up to two times harder than $SiO_2$ and $LiNbO_3$. The chemical composition of the film is measured via Raman spectroscopy[38] (see Fig. 2h). Analyzing the two C-C stretch modes ($D,G$) at 1312.4 $cm^{-1}$ and 1541.0 $cm^{-1}$, we find that our film contains up to 5% $sp_3$-hybridized C-atoms and a large hydrogen content, which classifies the material as amorphous diamond-like carbon (a-C:H), typical of PECVD deposition using methane precursor. We have carefully optimized the process to facilitate wafer-scale uniform growth of DLC for minimal stress and particle contamination, which is pivotal to the manufacture of photonic integrated waveguide

structures. The optical waveguide pattern is structured by DUV stepper photolithography (248 nm) and first transferred into a $Si_3N_4$ mask layer using standard fluoride-chemistry-based plasma etching. This intermediate mask has excellent resistance to oxygen plasma etching, which we use to transfer the waveguide pattern into the DLC hard mask (see Fig. 2c). With optimized argon ion beam etching (IBE), the etching selectivity between LN and DLC is up to ×3, which enables deep etching and steep sidewalls. Figure 1e contains an SEM cross section of the fully etched $LiNbO_3$ strip waveguide with waveguide base width of 3.3 μm and height of 0.7 μm featuring an 80° sidewall angle. Currently employed soft $SiO_2$-based masks are limited to an etch selectivity of ×1 in comparison. The film hardness directly correlates with the material etching rate in IBE, which is plotted as the etch selectivity compared to $LiNbO_3$ in Fig. 2g. Remarkably, our film hardness of 20 GPa lies at the

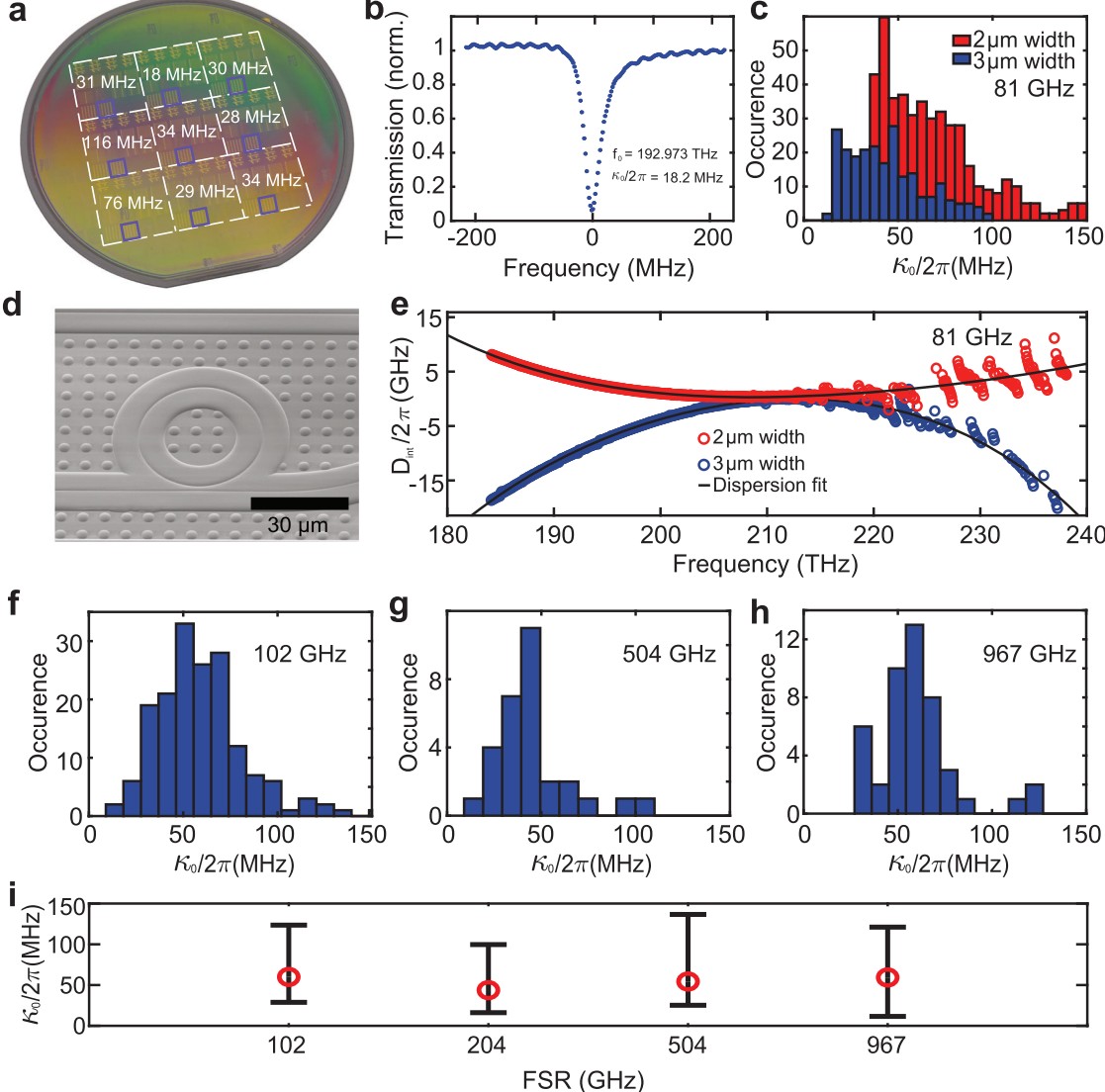

**Fig. 3 | Highly confining X-cut and Z-cut LiNbO₃ photonic integrated microresonator characterization. a** The most probable value $\kappa_0/2\pi$ of 9 chips measured on each of the fields shown on the 100 mm (4 in.) wafer. The reticle design contains 16 chips in each of the 9 fields uniformly exposed over the x-cut $LiNbO_3$ wafer. **b** Microresonator transmission (blue) and linewidth measurement of a single resonance at 192.97 THz from the racetrack resonator with 3 μm width waveguide using frequency-comb-assisted diode laser spectroscopy results in $\kappa_0/2\pi = 18.2$ MHz. **c** Histogram of $TE_{00}$ resonances from a 2 μm (red) and 3 μm (blue) wide single racetrack x-cut $LiNbO_3$ resonator showing the most probable value of $\kappa_0/2\pi = 45$ MHz. **d** SEM image of

microresonator with a radius of 20 μm from a z-cut $LiNbO_3$ wafer (corresponding to a ca. 1 THz FSR). **e** Measured integrated dispersion of the x-cut $LiNbO_3$ microresonator with free spectral range about 81 GHz, anomalous dispersion $D_2/2\pi = 105.72$ kHz for 2 μm waveguide width microresonator and normal dispersion of $D_2/2\pi = -291.23$ kHz for 3 μm waveguide width. Histograms of intrinsic cavity linewidths for $TE_{00}$ resonances of z-cut $LiNbO_3$ microresonators with FSRs of (**f**) 102 GHz, (**g**) 504 GHz, (**h**) 967 GHz. **i** Intrinsic cavity linewidth vs free spectral range of the microresonators. Red circles show the mean values from distributions in (**f, g, h**). Vertical bars show min and max values.

lower limit of the attainable hardness range of DLC thin films[38]. We remove the rough LiNbO$_3$ redeposition on the waveguide sidewall using SC-1 solution (NH$_4$OH: H$_2$O$_2$: H$_2$O=1:1:5) (see Fig. 2d) and the residual DLC mask (see Fig. 2d) using oxygen plasma to reveal the waveguide core (see Fig. 2e). The oxygen plasma from a photoresist asching tool is sufficient to completely remove the DLC without leaving any residues and this solution also causes no damage to the lithium niobate. Next, we fabricate the electrodes via a DUV-stepper lithography-based lift-off process for which we deposit 5 nm titanium adhesion layer and a 400 nm gold layer via electron-beam evaporation. Finally, the wafer is separated into chips via deep dry etching followed by backside grinding to obtain clean, vertical and smooth facets without a silicon pedestal for efficient waveguide edge coupling.

## High-density LiNbO$_3$ photonic integrated circuits characterization

Figure 3b depicts a normalized transmission (blue) of a resonance of 81 GHz FSR x-cut LiNbO$_3$ racetrack (chip D101 03 F3 C4) with 3 μm waveguide width and gold electrodes fabricated using our process and measured using frequency-comb-assisted diode laser spectroscopy[45].

The linewidth and dispersion measurement of the resonance at 192.97 THz yields an intrinsic photon loss rate $\kappa_0/2\pi = 18.2$ MHz, which corresponds to a quality factor of greater than 10 million and a linear propagation loss $\alpha = n_g/c \cdot \kappa_0$ of 4 dB m$^{-1}$, with the group velocity index $n_g = 2.275$ of our LiNbO$_3$ strip waveguide. This shows that our DLC hard mask approach yields ultra-low propagation loss. To verify that this performance is uniformly achieved across the wafer, we measure optical microresonators on each of the fields on the 100 mm wafer. Figure 3a shows the most probable value $\kappa_0/2\pi$ for TE modes of resonators on nine chips each located in one of the nine DUV lithography fields on the wafer. The measured chips are marked with blue squares. The data shows that the low propagation loss is indeed obtained. Figure 3e depicts the measured integrated dispersion $D_{int}/2\pi = \omega_\mu - \omega_0 - D_1/2\pi \cdot \mu$ of the LN microresonators with FSR $D_1/2\pi = 81$ GHz centered around $\omega_0/2\pi = 210$ THz. The anomalous GVD of $D_2/2\pi = 105.72$ kHz and normal GVD of $D_2/2\pi = -291.23$ kHz are measured for racetracks with waveguide width of 2 μm and 3 μm correspondingly. We observe mode mixing at wavelengths shorter than 1360 nm, which we attribute to the mixing of fundamental TE and TM modes in the ring resonator due to the inherent birefringence of the

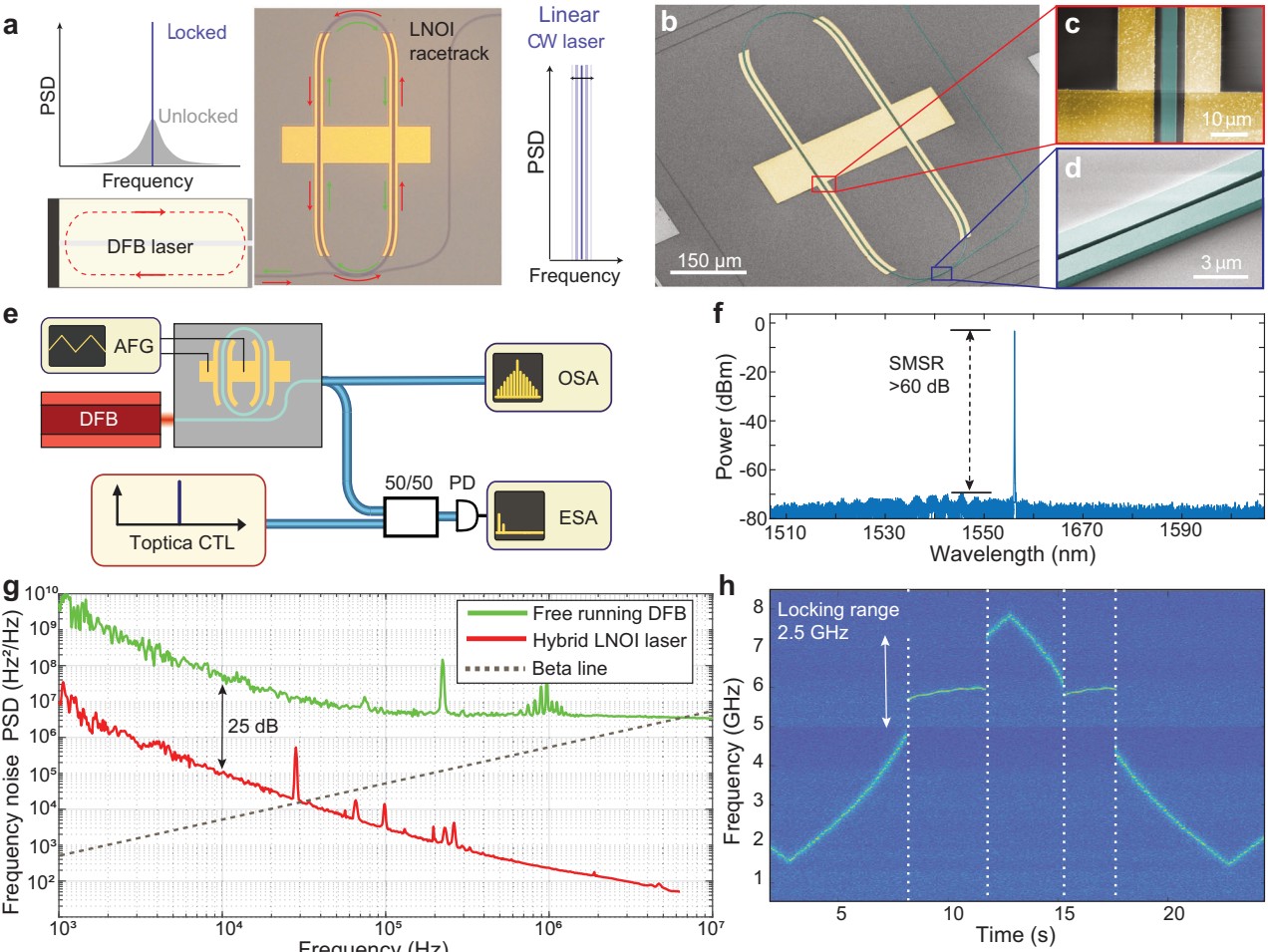

**Fig. 4 | III-V self-injection locked laser based on tightly confining LiNbO$_3$ photonic integrated circuits. a** Schematic of hybrid LNOI laser. DFB laser is self-injection locked to an LNOI photonic chip-based racetrack resonator. The bulk and surface scattering inside the LNOI waveguide induces back-scattered light and triggers the laser self-injection locking and frequency noise reduction. **b** SEM of the racetrack resonator with electrodes including high resolution insets of the waveguide with electrodes (**c**, red) and the coupling section (**d**, blue) (**e**) Schematic of the optical setup for laser characterization. Heterodyne beatnote measurement using free running Toptica CTL as a reference allows to measure laser frequency noise using a fast photodetector (FPD) and electrical spectrum analyzer (ESA). Optical spectrum analyzer (OSA). **f** Optical spectrum of the hybrid LNOI laser emission. The side mode suppression ratio (SMSR) is greater than 60 dB. **g** Single-sided PSD of frequency noise of the hybrid LNOI laser upon self-injection locking to racetrack resonator with 81 GHz FSR (red) and in free-running regime (green). **h** Spectrogram showing laser frequency change upon the linear tuning of the diode current, dashed areas correspond to the 2.5 GHz range where the laser is self-injection locked with minimal lasing frequency fluctuations and reduced laser linewidth.

material (see Supplementary Fig. 4). Next, we demonstrate the high areal density of our approach and fabricate record small LiNbO$_3$ photonic integrated resonators, that can be used in a variety of applications including filter resonators. It is important to emphasize that such microresonators would exhibit low Q-factor when using ridge waveguides, that exhibit leakage into the residual LiNbO$_3$ layer. Figure 3c, f, g, h shows histograms of intrinsic cavity linewidths for resonators with different FSRs: 81 GHz FSR x-cut LiNbO$_3$ race-track (D101_03_C04), 102 GHz FSR, 504 GHz FSR, 967 GHz FSR z-cut LiNbO$_3$ microrings correspondingly (chip D133_02_C15). Finally, we measured the mean intrinsic cavity linewidth of 55 MHz for a micro-resonator with 967 GHz FSR corresponding to a ring diameter of 40 μm (depicted in Fig. 3d), confirming the fabrication of high-density LN photonic integrated circuits.

## Ultrafast tunable low-noise laser based on tightly confining lithium niobate photonic integrated circuits

To demonstrate the utility of the platform, we demonstrate an MZM modulator (please refer to Supplementary Fig. 7) and a fast tunable low-noise laser, based on hybrid integration of a III-V/LNOI laser. We exploit the laser self-injection locking effect by directly edge-coupling an InP distributed feedback (DFB) laser with a LiNbO$_3$ racetrack reso-nator. The bulk and surface scattering inside the LiNbO$_3$ waveguide induces back-scattered light, provides spectrally narrowband fast optical feedback and triggers laser self-injection locking, and a reduction in the laser frequency noise power quadratically propor-tional to the loaded Q factor[46] according to $\frac{\delta\omega}{\delta\omega_{\text{free}}} \approx \frac{Q_{\text{DFB}}^2}{Q^2}\frac{1}{16R(1+\alpha_g^2)}$, where $\delta\omega_{\text{free}}/2\pi$ is the linewidth of the free-running laser; $\delta\omega/2\pi$ is the line-width of self-injection locked laser; $Q_{\text{DFB}}$ and $Q = \omega/\kappa$ are the loa-ded quality factors of the laser diode cavity and the microresonator mode and $\alpha_g$ is the phase-amplitude coupling factor (see the char-acterization of device used in Supplementary Fig. 5). The gap between the DFB and the LiNbO$_3$ chip is adjusted for the optimal optical

feedback phase, which entails equal locking range in forward and backward diode current scans. Figure 4d shows a spectrogram of the hybrid laser frequency change upon the linear tuning of the diode current, measured using heterodyne spectroscopy. The areas between dashed lines indicate the 2.5 GHz wide locking range, where the laser frequency is determined by the LiNbO$_3$ microresonator. Figure 4c shows that the frequency noise of the hybrid integrated self-injection locked laser is suppressed by more than 25 dB across the entire spectrum (see the full plot in Supplementary Fig. 6). The laser fre-quency noise reaches a value (white noise floor) of 52 Hz$^2$ Hz$^{-1}$ at 6 MHz offset. By integrating frequency noise power spectral density up to the frequency of the interception point with a line $S_\nu(f) = 8 \cdot \ln 2f/\pi^2$ (30 kHz for our device), we obtain a frequency measure $A$ for the full width at half maximum of the linewidth $\delta\nu = (8 \cdot \ln 2A)^{1/2}$[47]. Thus, the FWHM linewidth is 242 kHz at 1 ms integration time, 489 kHz at 10 ms, and 942 kHz at 100 ms. Using THz FSR resonators enabled by high-density integration advantage of our platform it should be possible to alleviate the need for using a DFB laser, as recently has been demon-strated with the RSOA gain chip being used for self-injection locking to Si$_3$N$_4$ chips with THz microrings[48]. The large FSR of the microresonator allows to achieve mode selection and laser self-injection locking with a single ring.

### Frequency-agile laser tuning

Next, we demonstrate the ability of fast frequency actuation of the laser using the Pockels effect of the LiNbO$_3$ microresonator. Figure 4d shows the maximal range over which the LiNbO$_3$ resonator frequency can be tuned while maintaining laser locked to the LiNbO$_3$ cavity. Figure 5b reveals one of the advantages of the LiNbO$_3$ platform in comparison to piezoelectric actuators—the modulation response function, measured using a vector network analyzer (see Fig. 5a), is flat up to the cavity cutoff frequency and no mechanical modes of the chip are excited[11]. In the self-injection locking range, the change of LiNbO$_3$

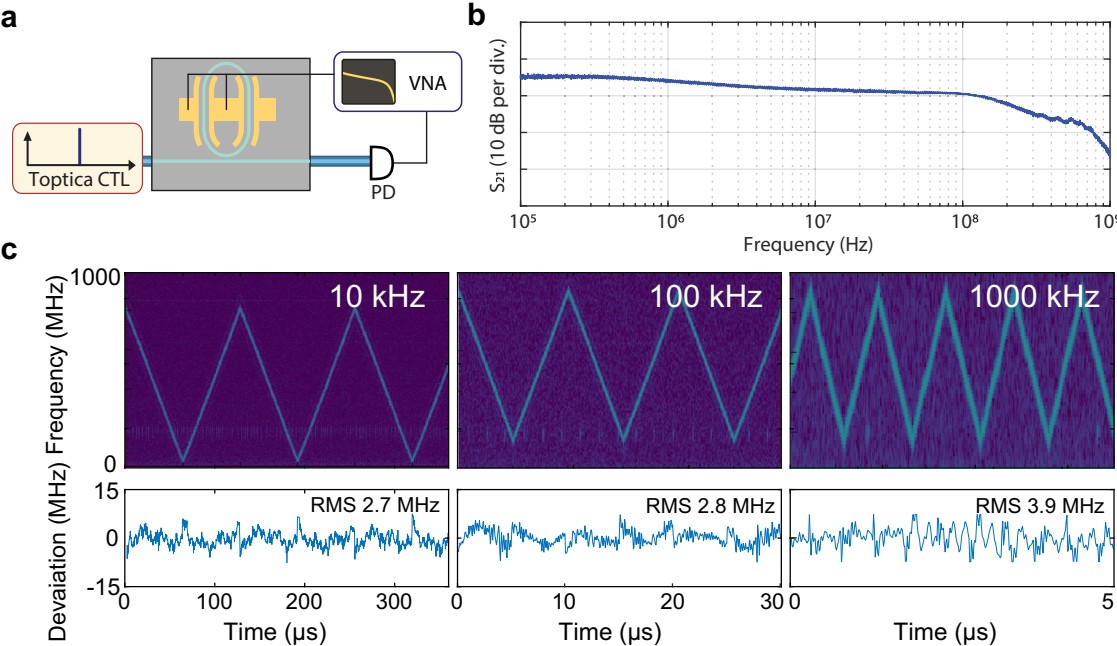

**Fig. 5 | Fast electro-optical tuning of a self-injection locked III-V LiNbO$_3$ based laser. a** Experimental setup for electro-optical response characterization of the LNOI microresonator laser using an external laser (Toptica CTL), a photoreceiver (PD) and a vector network analyzer (VNA). The beatnote is recorded on a fast oscilloscope (DSO) and analysed with short-time Fourier transforms. **b** Measured response of the electro-optic actuation for the 81 GHz FSR microresonator showing flat actuation bandwidth up to the cavity cutoff frequency. **c** Time-frequency spectrogram of the heterodyne beatnotes for triangular chirp repetition fre-quencies from 10 kHz to 1 MHz. The frequency excursion is 761 MHz at 10 kHz tuning rate, 730 MHz at 100 kHz and 721 MHz at 1 MHz chirp rate. Bottom row: residual of least-squares fitting of the time-frequency traces with symmetric tri-angular chirp pattern.

microresonator frequency directly changes the laser output frequency without additional feedback on the diode current. Figure 5c shows the main results of the heterodyne beat experiment with the DFB laser locked to a LiNbO$_3$ racetrack resonator. We define chirp nonlinearity as the root mean square (RMS) deviation of the measured frequency tuning curve from a perfect triangular ramp that is determined with least-squares fitting. Figure 5c presents the processed laser frequency spectrograms and the corresponding RMS nonlinearities upon applying to the electrodes triangular ramps with 2 V$_{p-p}$ amplitude at 10 kHz, 100 kHz, and 1 MHz frequencies. The measured tuning efficiency is 380 MHz V$^{-1}$ using only a single electrode on the side of the 81 GHz FSR racetrack resonator for self-injection locking. This value is more than tenfold larger than those achieved in AlN-based frequency agile lasers based on Si$_3$N$_4$ photonic integrated circuits and on par with those attained in PZT-actuated Si$_3$N$_4$ hybrid lasers[11] of 500 MHz V$^{-1}$.

At 1 MHz modulation frequency, the achieved RMS nonlinearity is as low as 3.9 MHz (relative nonlinearity 0.5%), which surpasses typical benchtop low-noise tunable external cavity diode laser systems. The tuning range of 760 MHz at high ramping speeds from 10 kHz up to 1 MHz, with small chirping RMS nonlinearities from 0.3% to 0.5% proves the frequency agility of our hybrid integrated laser. We also do not observe any tuning hysteresis or inherent nonlinear response of the LiNbO$_3$ microresonator.

## Discussion

In summary, we developed a platform for high-density lithium niobate on insulator photonic integrated circuits based on a deeply etched strip waveguide with tight optical confinement based on a micro structuring process featuring amorphous carbon films (DLC) as etch mask (see performance comparison of integrated lithium-niobate-based platforms in Supplementary Table 1). Our process is readily applied for a wide variety of photonic materials that are resistant to oxygen plasma etching, which includes silicon, silicon dioxide, silicon nitride and notably also ferroelectric oxides beyond LiNbO$_3$ such as BaTiO$_3$ or LiTaO$_3$. At similar loss levels compared to LiNbO$_3$-based ridge waveguides, our fully etched geometry affords four times smaller minimum bend radius, which corresponds to a potential increase of photonic component density by a factor of 16 without sacrificing significant electro-optical modulation efficiency, which is advantageous for applications in classical photonic and quantum computing with large switching fabrics or photonic networks[49]. In a similar manner, the increased optical confinement and small bend radii can greatly benefit other electro-optical quantum technologies such as quantum coherent microwave to optical converters[50,51]. Electro-optic materials can enable lasers with flat actuation bandwidth and very fast optical tuning up to multi-MHz bandwidth. Such a system is an attractive candidate for applications in coherent (FMCW) laser ranging and critically enables to reduce the voltage driving to CMOS level while achieving GHz tuning range, excellent linearity, no measurable hysteresis, and low laser phase noise.

## Data availability

Data used to produce the plots within this paper is available at https://doi.org/10.5281/zenodo.7966707. All other data used in this study are available from the corresponding authors upon request.

## Code availability

Code used to produce the plots within this paper is available at https://doi.org/10.5281/zenodo.7966707.

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

## Acknowledgements

This work was supported by funding by the EU Horizon Europe research and innovation program through grant no. 101113260 (HDLN). M.C. acknowledges funding from the EU H2020 Research and Innovation Program under grant agreement No. 812818 (MICROCOMB). A.S. acknowledges funding from European Space Technology Centre through ESA Contract No. 4000135357/21/NL/GLC/my. J.R. acknowledges funding from the SNSF through an Ambizione Fellowship with grant no. 201923. We acknowledge the contribution of Junqiu Liu in the design of testing structures in the early phase of the project. The samples were fabricated in the EPFL center of MicroNanoTechnology (CMi) and the Institute of Physics (IPHYS) cleanroom. The Raman spectrum was measured in the material characterization platform of IPHYS.

## Author contributions

Z.L., M.B., and R.N.W. fabricated the device with contributions from Z.T. T.J.K. proposed the use of DLC. J.R., J.Z., and M.C. simulated and designed the devices. J.Z., Z.L., J.R., G.L., and A.S characterized the devices. G.L., J.R., and N.K. performed laser characterization. Z.L., G.L., J.R., M.B., and M.C. prepared the figures and wrote the manuscript with input from all authors. T.J.K. supervised the project.

## Competing interests

The authors declare no competing financial interests. T.J.K. is a co-founder and shareholder of Luxtelligence SA, a foundry commercializing $LiNbO_3$ photonic integrated circuits, as well as DEEPLIGHT SA, a startup commercializing PIC-based frequency agile low noise lasers.
