## [Peer Review File · Nature Communications]

High density lithium niobate photonic integrated circuitsReviewer #1 (Remarks to the Author):

The manuscript by Li et al. reports using diamond-like carbon (DLC) as a new etch mask for the fabrication of lithium niobate on insulator (LNOI) photonic integrated circuits. LNOI is an emerging integrated photonics platform well suited for nonlinear and quantum applications. However, LN is difficult to etch, and how to make deeply etched LNOI waveguides with steep and smooth sidewalls remains an outstanding challenge. In this work, the authors used DLC as a hard mask and achieved over 3:1 etch selectivity as compared to the usual 1:1 ratio obtained using oxide-equivalent masks. The result of fully etched, 700 nm-thick waveguides, with 80-degree sidewall angle and 10 million Q-factor, is very impressive. The two extra experiments, namely hybrid laser integration and EO laser frequency tuning, are also interesting and nicely done.

Overall, I believe introducing DLC as a new mask for wafer-scale LNOI photonics fabrication is valuable to our community. This method may be extended to other "hard-to-etch" materials, and it has a broader impact in, for example, telecommunications and quantum applications.

Below are some detailed comments that I hope can help the authors further improve the manuscript.

1. The title states high density lithium niobate photonic integrated circuits and lasers. It's reasonable to say that the reported method can help increase waveguide density, but it's not clear how it could increase "laser" density. In this work, there is only one laser, and it's on a separate chip. I suggest the authors reconsider the title.
2. What's the radius of the rings being fabricated/measured in this paper? Based on Figure 1a and reported FSR, the ring radius is rather big. To support the claims of "high density", why don't the authors demonstrate rings with smaller radii and narrower widths (thus tighter confinement)?
3. What are the optical properties of DLC? Are they transparent at telecom/visible wavelengths?
4. The last two experiments (self-injection locking with III-V laser and EO frequency tuning) are somewhat disconnected from the main message of the paper. It's unclear how deeply etched LNOI waveguides are advantageous in these two applications as compared to traditional shallow-etched circuits. Can the authors clarify?
5. More details about the DLC removal process will be helpful. Is the O₂ plasma with or without bias? Does it affect LN? Also, is there any wet process that can more conveniently remove the DLC?
6. In multiple places, the authors extend the claims to "ferroelectric materials" in general. What are the other ferroelectric materials being referred to? Are they also having the same problem that no good reactive etching chemistry can be found and they have to rely on pure Ar etching?

Reviewer #2 (Remarks to the Author):

The authors demonstrate a technique for fully etching lithium niobate on insulator (LNOI). Lithium niobate is a material notably difficult to process, and state of the art techniques that result on low loss LNOI rely on strong ion bombardment, making it difficult to realize fully etched strip waveguide geometries. State of the art LNOI waveguides are ridge waveguides that have a relatively thick slab of lithium niobate underneath. This geometry features a stronger optical field confinement compared to traditional LN waveguides, but experiences higher bending losses than strip waveguide geometries commonly used in silicon photonics. As a result, dense photonic integration with tight bends and low coupling losses by edge coupling are not possible yet in LNOI.

The authors demonstrate that by using a hard mask of diamond-like carbon (DLC), which has very high selectivity compared to other traditional hard masks, it is then possible to withstand the aggressive Ar⁺ ion etching and fully etch the LN crystal all the way to the silica undercladding and

achieve strong confinement in the LN layer. The process comes with its own caveats, as the DLC itself is difficult to pattern and there is reminiscent roughness that is transferred onto the LN waveguide sidewalls. This is evidenced by the fact that the claimed ultralow loss level of 4dB/m is only achieved for wide (3 μ m) waveguides. Narrower waveguides exhibit Qs in the 1M range (supplementary Fig. 5).

The authors do not hide this fact and they do a very good job in presenting a statistical analysis of their devices. The manuscript is very clearly organized, succinct, and I particularly like that they include hybrid integration of lasers by self-injection locking to an electro-optic tunable high-Q resonators. These results constitute a very nice milestone in LNOI and in my view, the work deserves publication in Nature Communications.

I only have a couple of minor points for consideration:

1. Please clarify in the main text whether the LN crystal is X-cut or Z-cut. This becomes evident in the Supplementary information, but I think it is important to show in the main text.
2. What is the measured $V_{\pi L}$? Is there any clear advantage of the fabricated modulator compared to a ridge waveguide geometry?

Reviewer #3 (Remarks to the Author):

The authors demonstrate new LNOI processing to obtain low-loss tightly-confined waveguides and use their platform to demonstrate a fast tunable hybrid integrated laser. While these are results that exceed current state-of-the-art in specific aspects, they are limited in novelty. Additionally, in my opinion, the demonstration of the fast tunable hybrid lasers does not show an advantage of their newly developed platform compared to the already existing LNOI platforms.

The novelty of the manuscript seems to be separated into two parts:

- 1) a stronger confinement, without a clear demonstration benefiting from that confinement
- 2) an increased tuning rate by using an LNOI platform instead of a SiN platform, but not showing advantage of their platform over existing ones.

Part 1:

The authors demonstrate the use of diamond-like carbon (DLC) as a new hard mask for thin-film lithium niobate processing. Contrary to the state-of-the-art waveguides, they demonstrate fully-etched LNOI waveguides while keeping low losses, reporting 4 dB/m of propagation losses and Q-factors up to 10 million. However, rib waveguides have been shown to give similar losses and Q-factors (6 dB/m and 11 million, respectively)

[Desiatov, Boris, et al. "Ultra-low-loss integrated visible photonics using thin-film lithium niobate." *Optica* 6.3 (2019): 380-384.]

Other reported high-Q LNOI resonators are around 2 million:

[Lu, Juanjuan, et al. "Toward 1% single-photon anharmonicity with periodically poled lithium niobate microring resonators." *Optica* 7.12 (2020): 1654-1659.]

[McKenna, Timothy P., et al. "Ultra-low-power second-order nonlinear optics on a chip." *Nature Communications* 13.1 (2022): 4532.]

For that reason, the main novelty of the platform is the improved confinement. The authors nicely show that a reduction of about a factor 4 can be obtained in bending radius, while keeping losses the same. Indeed, this then gives rise to a significant improvement in integration density. Such reduction in two dimensions, comes down to a factor of 16, which is more than 1 order of magnitude improvement as claimed by the authors in the abstract. While correct, the limiting factor regarding integration density of x-cut LNOI racetrack modulators is often the length required for a CMOS-compatible V_{π} , and not the bending radius. Nevertheless, the authors indeed demonstrate a significant improvement in integration density.

Part 2:

In the second part of the manuscript, the authors demonstrate edge-coupling of an InP DFB laser to a fastly tuneable LNOI racetrack resonator in their platform. Through self-injection locking, they can lock the lasing wavelength to the resonator and obtain a reduction by more than 25 dB of the

frequency noise. Through electro-optic modulation of the racetrack resonator, they can tune the laser up to a speed of 1 MHz, over a range of 760 MHz.

The hybrid coupling of fabry-perot lasers to a ring resonator on the silicon nitride platform has been done before by different groups, including the authors themselves, reaching 20 dB reduction of the frequency noise:

[Siddharth, Anat, et al. "Near ultraviolet photonic integrated lasers based on silicon nitride." *Apl Photonics* 7.4 (2022): 046108.]

[Corato-Zanarella, Mateus, et al. "Widely tunable and narrow-linewidth chip-scale lasers from near-ultraviolet to near-infrared wavelengths." *Nature Photonics* (2022): 1-8.]

While the above literature is able to tune their lasers through integrated heaters, the novelty shown in this manuscript is the fast tuning through the electro-optic effect of lithium niobate. They reach faster tuning rates and therefore exceed the current state-of-the-art. However, this demonstration does not make use of any advantage their new platform gives over existing LNOI platforms, being the strong confinement. The Q factor of the used resonator is not mentioned in the manuscript, but as stated before, similar Q factors are obtained on existing LNOI platforms. Due to the limited novelty and a demonstration that is not showing an advantage over other platforms, the manuscript does not merit publication in *Nature Communications*.

A detailed response to all the reviews of the manuscript

We are grateful that three reviewers have seen our manuscript. After reading the comments made by the referees thoroughly, we would like to thank them for the detailed review and suggestions to improve the manuscript. We appreciate that the reviewers gave a positive evaluation such as:

“Overall, I believe introducing DLC as a new mask for wafer-scale LNOI photonics fabrication is valuable to our community.” (Reviewer #1)

“The manuscript is very clearly organized, succinct, and I particularly like that they include hybrid integration of lasers by self-injection locking to an electro-optic tunable high-Q resonators. These results constitute a very nice milestone in LNOI and in my view, the work deserves publication in Nature Communications.” (Reviewer #2)

“Nevertheless, the authors indeed demonstrate a significant improvement in integration density.” (Reviewer #3)

Before responding individually to each reviewer, we would like to first make an **Executive Response** to all reviewers to highlight the main novelty of our work:

We emphasize that our work's **key novelty** and **scientific significance** is a new fabrication process based on Diamond-Like Carbon (DLC) hard mask that allows for the first time to deeply etch LN with low loss and create thereby tightly confining Lithium Niobate strip waveguides, in contrast to the previous work that achieved and used only partial etching. The process is moreover DUV stepper based and thus scalable in volume. Tight confinement enables high density, i.e. tight bend radii, photonic integrated circuits, specifically a 16-fold increase in integration density.

In our revised manuscript, we demonstrate microresonators with up to 1 THz free spectral range corresponding to a ring diameter of 40 μm and measure mean intrinsic cavity linewidth of 55 MHz, confirming the fabrication of high-density LN photonic integrated circuits as requested by referees.

In addition following the request of referee #1, we have modified the title to “**High density lithium niobate photonic integrated circuits**”, and removed the connection to lasers.

We've also added to the SI the new V_{π} measurement as requested by referee #2 for a MZM modulator in x-cut LN fabricated with our process with 1.73 cm length and $V_{\pi}=1.94$ V.

We have addressed all the concerns and suggestions raised by the Referees and formulated our point-by-point response below. Below, we provide a point-by-point response to the reviewers' comments and questions. The **original review report is printed in black, our responses in blue, and the action taken in red**. In the revised manuscript, according to the reviewers' request, we have changed the title and added new data. We believe that these comments contribute to the paper's readability and clarity.

REVIEWER COMMENTS

Reviewer #1 (Remarks to the Author):

The manuscript by Li et al. reports using diamond-like carbon (DLC) as a new etch mask for the fabrication of lithium niobate on insulator (LNOI) photonic integrated circuits. LNOI is an emerging integrated photonics platform well suited for nonlinear and quantum applications. However, LN is difficult to etch, and how to make deeply etched LNOI waveguides with steep and smooth sidewalls remains an outstanding challenge. In this work, the authors used DLC as a hard mask and achieved over 3:1 etch selectivity as compared to the usual 1:1 ratio obtained using oxide-equivalent masks. The result of fully etched, 700 nm-thick waveguides, with 80-degree sidewall angle and 10 million Q-factor, is very impressive. The two extra experiments, namely hybrid laser integration and EO laser frequency tuning, are also interesting and nicely done.

Overall, I believe introducing DLC as a new mask for wafer-scale LNOI photonics fabrication is valuable to our community. This method may be extended to other “hard-to-etch” materials, and it has a broader impact in, for example, telecommunications and quantum applications.

We thank reviewer #1 for their careful study of our manuscript, constructive suggestions, and their positive assessment of the novelty and impact of our manuscript. We are happy to provide a detailed response to the raised questions below.

Below are some detailed comments that I hope can help the authors further improve the manuscript.

1. The title states high density lithium niobate photonic integrated circuits and lasers. It's reasonable to say that the reported method can help increase waveguide density, but it's not clear how it could increase “laser” density. In this work, there is only one laser, and it's on a separate chip. I suggest the authors reconsider the title.

We thank the reviewer for this suggestion, we removed ‘laser’ from the title of the paper.

Also, in our revised manuscript we add new data on the fabrication of microresonators using Z-cut lithium niobate with a radius of 20 μm (corresponding to ~ 1 THz FSR) - the smallest reported value for LN microresonators so far – thus further supporting our claim on high density PICs.

Action taken: We changed the title of our manuscript. We added new data on the fabrication and characterization of microresonators with 100 GHz, 200 GHz, 500 GHz and 1 THz FSRs.

2. What's the radius of the rings being fabricated/measured in this paper? Based on Figure 1a and reported FSR, the ring radius is rather big. To support the claims of “high density”, why don't the authors demonstrate rings with smaller radii and narrower widths (thus tighter confinement)?

We thank the reviewer for this comment. In our manuscript, we characterized devices with 81 GHz FSR. The racetrack resonators that were reported in the first version of the manuscript had an apex radius of 100 μm . In our revised manuscript we added new measurements of microrings with smaller radius and 100 GHz, 200 GHz, 500 GHz and 1 THz FSR, resulting in 40-55 MHz mean intrinsic cavity linewidth.

Figure 1 Photo of z-cut LNOI wafer with THz devices

Figure 2 (a) SEM image of cladded microring with the radius 20 μm . Histograms of intrinsic cavity linewidth for microresonators with (b) 102 GHz, (c) 504 GHz and (d) 0.97 THz FSRs. (e) Intrinsic cavity linewidth vs free spectral range of the microresonators. Red circles show the mean values from distributions in (b,c,d). Vertical bars show min and max values.

Action taken: We added new data on the characterization of microresonators with 100 GHz, 500 GHz and 1 THz FSRs.

3. What are the optical properties of DCL? Are they transparent at telecom/visible wavelengths?

We have not determined the optical properties of DLC at present. Because the DLC is removed without residue in the O_2 plasma etch step, it does not impact the performance and optical loss of the finished LNOI photonic components. According to the established literature [1], the transmission is up to 80-90% for 100nm DLC film at visible and up to 1000 nm wavelength.

[1] Boycheva, S., Popov, C., Kulisch, W., Bulir, J. & Piegari, A. Optical Properties of Nanocrystalline Diamond/Amorphous Carbon Composite Films. Fuller. Nanotub. Carbon Nanostructures 13, 457–469 (2005).

4. The last two experiments (self-injection locking with III-V laser and EO frequency tuning) are somewhat disconnected from the main message of the paper. It's unclear how deeply etched LNOI

waveguides are advantageous in these two applications as compared to traditional shallow-etched circuits. Can the authors clarify?

First of all, the two experiments (laser self-injection locking and EO frequency tuning) have been pursued to demonstrate the actual device based on our novel high-density LN photonic integrated. As such we do not consider them as disconnected but rather validating the approach.

Specifically, as far as self-injection locking is concerned, the latter requires high-quality (Q) resonators for efficient phase noise reduction of the free-running DFB diode and therefore makes inherent use of the platform's ability to achieve low loss (4 dB/m) as well as reasonable efficient input coupling. The frequency suppression ratio in laser self-injection locking for white noise is given by the formula: $\frac{\delta\omega}{\delta\omega_{free}} \sim \frac{Q_{DFB}^2}{Q_{SiN}^2}$

Concerning the specific advantages of deep etching we would like to emphasize, arguably in general, the 'gold standard' in wafer-scale manufacturing of PICs is to etch through the dielectric completely, which does not require timed etching and thus guarantees inherently uniformity across the wafer. Other advantages include:

- Less mode mixing due to the birefringence of LiNbO₃
- The absence of timed etching allows more precise thickness control
- DUV compatible process
- Steep sidewall angle
- Efficient input coupling due to the ability to fabricate thin inverse tapers
- Low optical loss
- High integration density

Deep waveguide etching technology has numerous advantages for photonic integrated circuit manufacture. Our deep etching technology facilitates the manufacture of much smaller microresonators for laser self-injection locking and Vernier-style external cavity diode lasers with bending radii that support THz FSR optical microresonators. For integrated Vernier lasers¹ this can **improve the tuning range** and single line operation due to the enlarged Vernier FSR calculated as $FSR_1 * FSR_2 / (FSR_1 - FSR_2)$, where FSR_1 , FSR_2 are free spectral ranges of two microrings used in Vernier ring filter. Indeed, in the recent demonstration² Vernier tuning range was limited by the large size of the optical LN microresonators.

Apart from the specific advantages of the deeply etched waveguide technology, the tight bending radius, **optical confinement, reduced phase error due to etch depth variation, and reduced mode mixing due to the vertical sidewalls**, allow to manufacture all other subcomponents of hybrid integrated lasers such as delay line spirals, splitters, with improved performance and fidelity. We also note that we can achieve this without compromising the optical propagation loss of the circuit. In particular, hybrid integrated lasers benefit from the very low optical loss of our microresonators, because the optical linewidth suppression factor scales with the square of the passive cavity loss rate.

Etch depth variations are particularly harmful in arrayed waveguide gratings, but also in Mach Zehnder structures where it requires heaters. To quantify this error it is illustrative to consider the phase error,

i.e. $\frac{\partial n_{eff}}{\partial h_{slab}}$, where h_{slab} is a height of the slab.

To illustrate the **reduced phase error** advantage in fully etched waveguides we perform numerical simulations. Figure 2 of this reply shows the dependence of the effective index (blue trace) and the phase error (red trace) as a function of the etching depth. Effective refractive index n_{eff} becomes more sensitive to the etching depth, the thicker the slab becomes. The simulation was carried out for a $2\ \mu\text{m}$ x $0.6\ \mu\text{m}$ waveguide with 77 degrees of etch angle.

Figure 3. Dependence of the effective index (blue) and the phase error (red) as function of the etching depth. Inset shows cross-section of the LN waveguide used for simulations.

5. More details about the DLC removal process will be helpful. Is the O_2 plasma with or without bias? Does it affect LN? Also, is there any wet process that can more conveniently remove the DLC?

The oxygen plasma from a photoresist ashing tool is sufficient to completely remove the DLC without leaving any residues. And this solution also causes no visible damage to the lithium niobate. Therefore, it is not necessary to apply bias. Applying bias in an RIE tool introduces a stronger physical component to the removal process, which could roughen the sidewalls of the lithium niobate devices, especially near their top surface.

The DLC removal in the standard photoresist ashing tool has been so convenient, and reliable and complete that there was no need to explore other wet etching solutions to remove the mask. In fact, we do believe that this is a key advantage of our process; in contrast to e.g. SiO_2 that requires wet etching to remove the mask. In this former case, we would also need to make sure the wet etching does not damage lithium niobate.

Action taken: We added as requested by the referee details on how the DLC is removed to the main manuscript.

6. In multiple places, the authors extend the claims to “ferroelectric materials” in general. What are the other ferroelectric materials being referred to? Are they also having the same problem that no good reactive etching chemistry can be found and they have to rely on pure Ar etching?

Yes, other ferroelectric materials have the same problem³. We refer to e.g. LiTaO_3 , BTO, KNbO_3 . It is important to note that dry etching recipes do exist even for LiNbO_3 however the fact that the compound formed is not volatile⁴ and other chemical structure-related effects of the perovskites, compound the etch quality and requires further processing steps. For Ar etching which broadly applies to all ferroelectrical materials, a suitable hardmask needs to be found with sufficient selectivity, which is where DLC stands out (in contrast to SiO_2).

We are trying the fabrication of photonic integrated circuits with microresonators using some of the above-mentioned materials, but it goes beyond the scope of this paper.

Action taken: We expand the discussion on ferroelectric materials in the discussion section of the main manuscript.

Reviewer #2 (Remarks to the Author):

The authors demonstrate a technique for fully etching lithium niobate on insulator (LNOI). Lithium niobate is a material notably difficult to process, and state of the art techniques that result on low loss LNOI rely on strong ion bombardment, making it difficult to realize fully etched strip waveguide geometries. State of the art LNOI waveguides are ridge waveguides that have a relatively thick slab of lithium niobate underneath. This geometry features a stronger optical field confinement compared to traditional LN waveguides, but experiences higher bending losses than strip waveguide geometries commonly used in silicon photonics. As a result, dense photonic integration with tight bends and low coupling losses by edge coupling are not possible yet in LNOI.

We thank the reviewer for their support of our manuscript and the acknowledgement of the novelty of our fabrication and LNOI device technology.

The authors demonstrate that by using a hard mask of diamond-like carbon (DLC), which has very high selectivity compared to other traditional hard masks, it is then possible to withstand the aggressive Ar⁺ ion etching and fully etch the LN crystal all the way to the silica undercladding and achieve strong confinement in the LN layer. The process comes with its own caveats, as the DLC itself is difficult to pattern and there is reminiscent roughness that is transferred onto the LN waveguide sidewalls. This is evidenced by the fact that the claimed ultralow loss level of 4dB/m is only achieved for wide (3μm) waveguides. Narrower waveguides exhibit Qs in the 1M range (supplementary Fig. 5).

Part of the beauty of this DLC hardmask process is that while very resistant to physical etching, DLC is very easily etched with oxygen ions. As detailed in the SI, a thin PECVD silicon nitride layer is deposited on the DLC. The device designs are first etched into this silicon nitride layer by fluoride-based RIE, and then this thin layer is sufficient to serve as an etch mask for the DLC etch by an oxygen-based RIE process. This bilayer hardmask approach makes the DLC very easy to pattern.

As for the reminiscent roughness, we completely agree with the reviewer. The roughness is visible in Figure 2c of the main manuscript, and the inset of this figure shows that there is even a slight undercutting of the DLC below the silicon nitride. We also provide an additional SEM image (see Figure 4 of this reply) of the DLC mask which is close to the cleavage facet. The SEM shows that there is an unconsolidated thin layer on the surface of the DLC. The roughness of this layer doesn't have an obvious influence on the LN etching. And below this layer, is the dense DLC. We are in the process of further optimizing the DLC etching recipe in the ICP-RIE tool that we are using, playing with process pressure and substrate bias to get more pronounced etch anisotropy. Thus, we hope to reduce the roughness of the DLC hardmask sidewalls in the near future.

Figure 4. SEM of the DLC mask which is close to the cleavage facet. The SEM shows that there is an unconsolidated thin layer on the surface of the DLC. The roughness of this layer doesn't have an obvious influence on the LN etching. And below this layer, is the dense DLC.

The authors do not hide this fact and they do a very good job in presenting a statistical analysis of their devices. The manuscript is very clearly organized, succinct, and I particularly like that they include hybrid integration of lasers by self-injection locking to an electro-optic tunable high-Q resonators. These results constitute a very nice milestone in LNOI and in my view, the work deserves publication in Nature Communications.

We thank the reviewer for the positive evaluation of our results.

I only have a couple of minor points for consideration:

1. Please clarify in the main text whether the LN crystal is X-cut or Z-cut. This becomes evident in the Supplementary information, but I think it is important to show in the main text.

We thank the reviewer for noting this omission, we used X-Cut LNOI crystal. We clarify this in the main text.

2. What is the measured $V_{\pi L}$? Is there any clear advantage of the fabricated modulator compared to a ridge waveguide geometry?

In the first version of the manuscript, we did not present EO modulators. Here we add measurements (see Figure 5 of this reply) of an MZM modulator with a 1.73 cm length and $V_{\pi} = 1.94$ V, resulting in $V_{\pi} L = 3.3$ V×cm MZM modulator with the push-pull configuration that has an LN waveguide width of 1.5 μ m and an LN waveguide–electrode gap of 2 μ m.

Figure 5 (a) Photo of 1.73 cm long MZM modulators. (b) Photo of the 100 mm x-cut LN wafer. (c) V_{π} measurement for a MZM modulator with a 1.73 cm length and $V_{\pi} = 1.94$ V.

As discussed in the main text and presented in Figure 1 (panel I), there is no clear advantage, but also no clear disadvantages of using slab vs. ridge waveguide geometries for electrooptic modulators, as the efficiency is given by the r_{33} material constant. At small inter-electrode distances, both platforms perform with the same efficiency. However, if the electrode gap is increased (to avoid any metal-induced loss), the ridge waveguide platform starts to overcome the strip waveguide in terms of electro-

optic efficiency. Thus, there is a tradeoff - if you tolerate losses you can make low $V_{\pi}L$. This is important for a fair comparison with other works.

One advantage of tighter bends is the ability to put more devices on a chip, e.g. by using meander waveguides. This may be of particular interest to e.g. creating modulators with exceptionally low V_{π} as used for cryogenic interconnects for superconducting circuits. Another advantage of our fabrication method is more efficient input coupling due to the ability to fabricate inverse tapers with a small taper width. Another advantage is that we have reliably low loss using our approach (see Table 1 for comparison).

Other advantages might include the realization of polarization splitters on an LN chip with 400-500 nm thickness and 2-3 μm waveguide width, by mixing TE_{00} and TM_{10} , however, this demonstration is beyond the scope of our work.

Action taken: We added new data on the fabrication and characterization of $V_{\pi}L$ of MZM modulator in the SI.

Reviewer #3 (Remarks to the Author):

The authors demonstrate new LNOI processing to obtain low-loss tightly-confined waveguides and use their platform to demonstrate a fast tunable hybrid integrated laser. While these are results that exceed current state-of-the-art in specific aspects, they are limited in novelty. Additionally, in my opinion, the demonstration of the fast tunable hybrid lasers does not show an advantage of their newly developed platform compared to the already existing LNOI platforms.

The novelty of the manuscript seems to be separated into two parts:

- 1) a stronger confinement, without a clear demonstration benefiting from that confinement
- 2) an increased tuning rate by using an LNOI platform instead of a SiN platform, but not showing advantage of their platform over existing ones.

We thank the reviewer for the appreciation of the platform and the result on improved integration density.

Part 1:

The authors demonstrate the use of diamond-like carbon (DLC) as a new hard mask for thin-film lithium niobate processing. **Contrary to the state-of-the-art waveguides, they demonstrate fully-etched LNOI waveguides while keeping low losses, reporting 4 dB/m of propagation losses and Q-factors up to 10 million. However, rib waveguides have been shown to give similar losses and Q-factors (6 dB/m and 11 million, respectively)**

[Desiatov, Boris, et al. "Ultra-low-loss integrated visible photonics using thin-film lithium niobate." *Optica* 6.3 (2019): 380-384.]

Other reported high-Q LNOI resonators are around 2 million:

[Lu, Juanjuan, et al. "Toward 1% single-photon anharmonicity with periodically poled lithium niobate microring resonators." *Optica* 7.12 (2020): 1654-1659.]

[McKenna, Timothy P., et al. "Ultra-low-power second-order nonlinear optics on a chip." *Nature Communications* 13.1 (2022): 4532.]

We respectfully disagree with the reviewer and judge the novelty of our fabrication process and performance highly. We would like to point out that we achieve not only low optical losses but **critically have high yield and reproducibility across a 100 mm wafer**, on devices manufactured **using DUV stepper lithography** – a wafer process, that is industry standard (in contrast to the academic results that use e-beam).

We offer below a careful comparison of our work to the state of the art reported in the literature.

The work of [Desiatov, Boris, et al. "Ultra-low-loss integrated visible photonics using thin-film lithium niobate." *Optica* 6.3 (2019): 380-384.] **concerns the visible spectral range**, which for the same photon lifetime has a higher $Q=\tau\omega$. Therefore the comparison cannot be made directly to our 1550 nm result. Moreover, the work **does not present any statistics** and also employed a wide waveguide (compared to the wavelength) reducing optical losses. It is therefore not clear if the authors are sampling the wings of the Q-factor distribution, and reported an outlier. In fact, despite being 4 years old and a very active community on LNOI, the work has since then not been reproduced by any other group. Assuming that sidewall scattering is the dominant loss factor and spectral dependence of the loss according to Rayleigh-like scattering (k^4), we would expect a quality factor of $(1550/635)^3 \times 11e6 = 159$ M and a propagation loss around 0.1 dB/m in the Harvard LNOI platform at 1550 nm. This is more than one order of magnitude below the state-of-the-art for lithium niobate achieved either at Harvard or anywhere else at 1550 nm.

Moreover, the original report from the Loncar group at 1550 nm [M. Zhang et al., “*Monolithic ultra-high-Q lithium niobate microring resonator*,” *Optica*, 2017] has a loss of 38 MHz of an individual isolated resonance. For all recent papers [refs 2,3,11,12 from Table 1] (including the same group) the losses reported are from 45 MHz to 180 MHz corresponding to 8 to 30 dB/m (which are fabricated using EBL, not stepper lithography). The same comment on the statistics applies - the report concerns isolated resonances, but it is not clear to what extent high Q-factors for all resonances can be attained. Hence it may be a sampling of the statistics in the wings of the distribution.

Lastly, only a few published results on ridge waveguides using scalable photolithography such as DUV stepper lithography. In particular, the publication by K. Luke et al.[5], the best achieved losses using DUV stepper lithography and silicon oxide hard mask technology were **only 0.21 dB/cm with a median loss of 0.27 dB/cm**. Compared to Luke et al., we achieve a best loss of 0.04 dB/cm and a median loss of 0.09 dB/cm across the whole wafer.

Taken together, our approach does achieve a loss level that is comparable to the best results and, importantly, includes a statistical analysis and quotes **not individual resonance** but the **mean of the distribution**. We firmly believe that this is a more appropriate way to quote loss than reporting individual data points.

Action taken: We put a comparison table of lithium niobate platforms to the SI.

For that reason, the main novelty of the platform is the improved confinement. The authors nicely show that a reduction of about a factor 4 can be obtained in bending radius, while keeping losses the same. Indeed, this then gives rise to a significant improvement in integration density. Such reduction in two dimensions, comes down to a factor of 16, which is more than 1 order of magnitude improvement as claimed by the authors in the abstract. While correct, the limiting factor regarding integration density of x-cut LNOI racetrack modulators is often the length required for a CMOS-compatible V_π, and not the bending radius. Nevertheless, the authors indeed demonstrate a significant improvement in integration density.

The improved confinement is a result of a fabrication process we developed. We present a comparison to other works in Table 1. We also achieved higher input coupling efficiency, we used a DUV stepper, not EBL; in the revised version we also demonstrate 1 THz FSR LN microresonators for the first time.

In a revised manuscript, we show devices with 100 GHz, 200 GHz 500 GHz, and 1 THz FSR in Z-cut LN and demonstrate the mean intrinsic linewidth of 50 MHz.

Ref	Fabrication	Intrinsic Q-factors	Linear optical loss	Statistical analysis	V _π L product E-O efficiency
This Work	DUV stepper + DLC	≈10⁷	0.04 dB/cm	Yes	3.9 V cm (MZM)
[1] ⁵	EBL	≈10 ⁷	0.03 dB/cm	No	No data
[2] ²	EBL	1.2×10 ⁶	0.3 dB/cm	No	~300 MHz V ⁻¹
[3] ⁶	EBL	2.5×10 ⁶	No data	No	500 MHz V ⁻¹
[4] ^{7*}	EBL	7.8×10 ⁶	0.06 dB/cm	No	1.6 V cm
[5] ⁸	DUV stepper	1.8×10 ⁶	0.27 dB/cm	Yes	No data
[6] ⁹ , [7] ¹⁰	Laser writing + CMP	7.1×10 ⁶	No data	No	2.16 V cm
[8] ¹¹	EBL	No data	No data	No	2.5 V cm
[9] ¹²	EBL	No data	No data	No	2.15 V cm
[10] ¹³	EBL	No data	0.41 dB/cm	No	No data
[11] ¹⁴	EBL	~4×10 ⁶	No data	No	No data
[12] ¹⁵	EBL	~1×10 ⁶	No data	No	No data

* Loss measurement was done at visible wavelength.

- [1] M. Zhang et al., “*Monolithic ultra-high-Q lithium niobate microring resonator*,” *Optica*, 2017
- [2] M. Li et al., “*Integrated Pockels laser*”. *Nat Commun* 13, 5344 (2022).
- [3] M. Zhang et al., “*Electronically programmable photonic molecule*,” *Nature Photonics*, 2019
- [4] B. Desiatov et al. "Ultra-low-loss integrated visible photonics using thin-film lithium niobate", *Optica* 6.3 (2019): 380-384.
- [5] K. Luke et al., “*Wafer-scale low-loss lithium niobate photonic integrated circuits*,” *Optics Express*, 2020
- [6] Z. Fang et al. "Efficient electro-optical tuning of an optical frequency microcomb on a monolithically integrated high-Q lithium niobate microdisk", *Opt. Lett.* 44, 5953-5956 (2019)
- [7] R. Wu et al. “*High-Production-Rate Fabrication of Low-Loss Lithium Niobate Electro-Optic Modulators Using Photolithography Assisted Chemo-Mechanical Etching (PLACE)*”, *Micromachines*, 13, 378 (2022).
- [8] Mengyue Xu et al. “*High-performance coherent optical modulators based on thin-film lithium niobate platform*”, *Nat Commun* 11, 3911 (2020).
- [9] A. Shams-Ansari et al, "Electrically pumped laser transmitter integrated on thin-film lithium niobate", *Optica* 9, 408-411 (2022)
- [10] I. Krasnokutska et al, "Ultra-low loss photonic circuits in lithium niobate on insulator", *Opt. Express* 26, 897-904 (2018)
- [11] Y. He et al, "Self-starting bi-chromatic LiNbO3 soliton microcomb", *Optica* 6, 1138-1144 (2019)
- [12] Z. Gong et al, "Near-octave lithium niobate soliton microcomb," *Optica* 7, 1275-1278 (2020)

Even though we agree with the Referee that the Q-factors achieved in our platform do not exceed the highest reported ones, we believe that our level of statistical analysis surpasses the prior works on lithium niobate platforms. We attach here a table of comparison with other works on integrated lithium niobate, where we highlight the fact that the record achieved Q-factors had no statistical proof. The actual reported⁸ **wafer-scale** losses are higher by an order of magnitude (0.27 dB/cm) than those presented for single resonances. We provide detailed information on optical losses at different locations on the wafer and for different devices. Moreover, we find the comparison with work⁷ of Desiatov’s incorrect, as the Q-factors presented there were obtained for the visible light.

In summary, there are several important parameters (loss, $V_{\pi}L$, waveguide dimensions, operational wavelength, wafer scale) that should be considered for a fair comparison with other works.

Action taken: We added to the SI a table of comparison for LNOI resonators and modulators demonstrations.

Part 2: In the second part of the manuscript, the authors demonstrate edge-coupling of an InP DFB laser to a fastly tuneable LNOI racetrack resonator in their platform. Through self-injection locking, they can lock the lasing wavelength to the resonator and obtain a reduction by more than 25 dB of the frequency noise. Through electro-optic modulation of the racetrack resonator, they can tune the laser up to a speed of 1 MHz, over a range of 760 MHz. The hybrid coupling of fabry-perot lasers to a ring resonator on the silicon nitride platform has been done before by different groups, including the authors themselves, reaching 20 dB reduction of the frequency noise:

[Siddharth, Anat, et al. "Near ultraviolet photonic integrated lasers based on silicon nitride." *Apl Photonics* 7.4 (2022): 046108.]

[Corato-Zanarella, Mateus, et al. "Widely tunable and narrow-linewidth chip-scale lasers from near-ultraviolet to near-infrared wavelengths." Nature Photonics (2022): 1-8.]

While the above literature is able to tune their lasers through integrated heaters, the novelty shown in this manuscript is the fast tuning through the electro-optic effect of lithium niobate. They reach faster tuning rates and therefore exceed the current state-of-the-art. However, this demonstration does not make use of any advantage their new platform gives over existing LNOI platforms, being the strong confinement. The Q factor of the used resonator is not mentioned in the manuscript, but as stated before, similar Q factors are obtained on existing LNOI platforms.

We thank the reviewer for this comment, and for providing the comparison to other laser demonstrations. First, we do not agree with this comment as the Q factor of the microresonator used in the laser experiment (81 GHz FSR) was mentioned in the SI.

For the laser application, the advantages of tight confining LN waveguides are the following. First, for the laser self-injection locking approach, using THz rings might alleviate the need of using DFB lasers. Recently, the RSOA gain chip was used for self-injection locking to the Si₃N₄ chip with the THz device¹⁶. Large FSR of microresonator allowed to achieve mode selection and laser self-injection locking without using DFB, DBR or Fabry-Perot type of laser. Second, an external cavity laser with a Vernier ring filter implemented in tightly confining LN PIC with THz rings can improve the tuning range and single-line operation due to the enlarged Vernier FSR. Third, fully etched waveguides facilitate the design and fabrication of bus waveguide-ring coupling allowing overcoupled microresonators for Vernier ring filter-based lasers².

Action taken: We add a discussion of the advantage of tight confinement waveguides for SIL lasers in the main text.

Due to the limited novelty and a demonstration that is not showing an advantage over other platforms, the manuscript does not merit publication in Nature Communications.

We respectfully disagree, as our fabrication process exploits DUV stepper lithography, compared to EBL used in most previous works (see the comparison table) and introduces DLC hard mask. Our process allows us to reliably achieve high Q factors among many devices on the wafer, which is confirmed by statistical analysis of two fabrication runs (x-cut and z-cut LN). In our opinion, a fair comparison can be done only with the HyperLight results⁸ claiming the best loss of 0.21 dB/cm, as they avoid EBL.

References:

1. Tran, M. A., Huang, D. & Bowers, J. E. Tutorial on narrow linewidth tunable semiconductor lasers using Si/III-V heterogeneous integration. *APL Photonics* **4**, 111101 (2019).
2. Integrated Pockels laser | Nature Communications. <https://www.nature.com/articles/s41467-022-33101-6>.
3. Luo, Z., Shao, S. & Wu, T. Characterization of AlN and AlScN film ICP etching for micro/nano fabrication. *Microelectron. Eng.* **242–243**, 111530 (2021).
4. Shen, C. *et al.* A comparative study of dry-etching nanophotonic devices on a LiNbO₃-on-insulator material platform. in *4th Optics Young Scientist Summit (OYSS 2020)* vol. 11781 212–217 (SPIE, 2021).
5. Zhang, M., Wang, C., Cheng, R., Shams-Ansari, A. & Lončar, M. Monolithic ultra-high-Q lithium niobate microring resonator. *Optica* **4**, 1536 (2017).
6. Zhang, M. *et al.* Electronically programmable photonic molecule. *Nat. Photonics* **13**, 36–40 (2019).
7. Desiatov, B., Shams-Ansari, A., Zhang, M., Wang, C. & Lončar, M. Ultra-low-loss integrated visible photonics using thin-film lithium niobate. *Optica* **6**, 380 (2019).
8. Luke, K. *et al.* Wafer-scale low-loss lithium niobate photonic integrated circuits. *Opt. Express* **28**, 24452 (2020).
9. Efficient electro-optical tuning of an optical frequency microcomb on a monolithically integrated high-Q lithium niobate microdisk. <https://opg.optica.org/ol/fulltext.cfm?uri=ol-44-24-5953&id=423888>.
10. Wu, R. *et al.* High-Production-Rate Fabrication of Low-Loss Lithium Niobate Electro-Optic Modulators Using Photolithography Assisted Chemo-Mechanical Etching (PLACE). *Micromachines* **13**, 378 (2022).
11. Xu, M. *et al.* High-performance coherent optical modulators based on thin-film lithium niobate platform. *Nat. Commun.* **11**, 3911 (2020).
12. Shams-Ansari, A. *et al.* Electrically pumped laser transmitter integrated on thin-film lithium niobate. *Optica* **9**, 408–411 (2022).
13. Krasnokutskaya, I., Tambasco, J.-L. J., Li, X. & Peruzzo, A. Ultra-low loss photonic circuits in lithium niobate on insulator. *Opt. Express* **26**, 897 (2018).
14. He, Y. *et al.* Self-starting bi-chromatic LiNbO₃ soliton microcomb. *Optica* **6**, 1138 (2019).
15. Gong, Z., Liu, X., Xu, Y. & Tang, H. X. Near-octave lithium niobate soliton microcomb. *Optica* **7**, 1275 (2020).
16. Corato-Zanarella, M. *et al.* Widely tunable and narrow-linewidth chip-scale lasers from near-ultraviolet to near-infrared wavelengths. *Nat. Photonics* **17**, 157–164 (2023).

Reviewer #1 (Remarks to the Author):

The authors have fully addressed my previous review comments, and I support its publication in Nature Communications.

Reviewer #2 (Remarks to the Author):

The authors have done a very careful job in addressing the reviewers' comments. I recommend the publication of the manuscript in its present form in Nature Communications.

Reviewer #3 (Remarks to the Author):

Dear editor and authors,

The author's have made a compelling case in which they have responded well to all my comments. I have no further comments

Best

A response to all the reviews of the manuscript

We thank the reviewers for the positive evaluation of our revised manuscript.